# Extraction of Phenolic Compounds from *Tabernaemontana catharinensis* Leaves and Their Effect on Oxidative Stress Markers in Diabetic Rats

**DOI:** 10.3390/molecules25102391

**Published:** 2020-05-21

**Authors:** Rafael Sari, Paula Conterno, Leticia Dangui da Silva, Vanderlei Aparecido de Lima, Tatiane Luiza Cadorin Oldoni, Gustavo Roberto Thomé, Solange Teresinha Carpes

**Affiliations:** Department of Chemistry, Federal University of Technology—Paraná (UTFPR), P.O. Box 591, 85503-390 Pato Branco, Brazil; rafael.mazzonetto@gmail.com (R.S.); paulabioconterno@gmail.com (P.C.); ltcdangui@gmail.com (L.D.d.S.); valima66@gmail.com (V.A.d.L.); tatianeoldoni@utfpr.edu.br (T.L.C.O.); gurotho@gmail.com (G.R.T.)

**Keywords:** phenolic compound, high performance liquid chromatography, diabetes, superoxide dismutase, catalase activity, *Tabernaemontana catharinensis*

## Abstract

The aim of this study was to evaluate the most effective extraction condition (temperature, solvent type and time) for recovery of high-value phytochemicals present in the *Tabernaemontana catharinensis* leaves (TC) and to assess their effect on biochemical parameters in streptozotocin-induced diabetic rats. The extraction of phenolic compounds from TC using a factorial design (FD) 2³, high performance liquid chromatography (HPLC), response surface methodology (RSM) and principal component analysis (PCA) were studied. It was found that the optimal conditions for extraction of phenolics were higher temperature (65 °C) and time (60 min) using ethanol as extractor solvent. In this condition of extraction (A8), total phenolic compounds (TPC) and antioxidant activity (AA) were determined. Additionally, this extract was used to evaluate their effect on antioxidant enzyme activities (superoxide dismutase (SOD) and catalase (CAT)) as well as lipid peroxidation (LP) and protein thiols level (PSH) in the liver and kidneys of normal and diabetic rats. As result, *T. catharinensis* extract presented TPC content of 23.34 mg EAG/g (equivalent gallic acid) and AA of 34.26 μmol Trolox/g. Phenolic acids (ferulic acid and coumaric acid) and flavonoids (quercetin, rutin and pinocembrin) could be recovered and identified by HPLC. This study indicated an important role of the *T. catharinensis* extract on free radical inactivation and on the antioxidant defense system in diabetic rats. In fact, the use of *T. catharinensis* extract restored the normal activity of SOD (*p* < 0.05) and suppressed malondialdehyde levels in liver and kidney tissues. Thus, the *T. catharinensis* extract, rich in phenolic compounds, can be responsible for the recover the enzymatic changes in the liver and kidney tissues provoked by diabetes in rats. In addition, the lipid peroxidation rate decreased in the diabetic rats treated with *T. catharinensis*.

## 1. Introduction

The use of plants by humans for nutritional and medicinal purposes has been historically reported [1,2,3]. The importance of natural products is clearly recognized [4,5] and their challenges include extraction [6] and identification of chemical composition [2,6]. In addition, the potential applications of these bioactive compounds need to be consolidated in diverse areas such as in food [7,8,9], chemical and pharmaceutical [4,5,6] industries.

In fact, nature, accompanied by its great variability of species, consists of an expressive source of chemical compounds known as secondary metabolites with great potential in the development of new drugs [2,6,10]. Phenolics are phytochemical compounds, which present themselves as the most important group in natural products [11]. The phenolic compounds act as antioxidant and can be effective in preventing free radical formation by scavenging them and thus suppressing the oxidative stress, which is responsible for the induction of many chronic and degenerative diseases [1,4,11,12,13].

Plants and their compounds can be used for ethnopharmacological studies in the search for new bioactive compounds that produce therapeutic action in pathologies such as hypertension [2] and diabetes [3]. In fact, diabetes-related diseases may affect more than 500 million people by 2030 and currently is the seventh leading cause of mortality globally [12]. Diabetes leads to changes in lipid and glucose metabolism, as well as changes in enzyme levels, which may contribute to long-term tissue damage [13]. Currently, diabetes is being treated with herbal medicines associated with conventional therapies. In this sense, several plant-based pharmacologically active ingredients used traditionally in the world have shown anti-diabetic property, such as *Ferula assa-foetida* [13], *Catharanthus roseus* [14] and *Syzygium mundagam* [15]. These plants are gaining popularity due to their natural origin and relatively lesser side effects.

In Brazil, the great biodiversity reflects the richness of its flora and provide the stimulus to study and identify new bioactive compounds. *Tabernaemontana catharinensis* is one of the 3700 species that comprises the family Apocynaceae [16] and is widely distributed in Brazil, Argentina and Paraguay. This plant is popularly known as “cobrina”, “jasmim” or “casca de cobra” (snake skin) [17]. The main features of the species are small shrub, elliptic leaves to ovate-elliptic, inflorescence and with whitish flowers [18]. Leaves, bark, seeds and roots of this plant are known for its use in popular medicine due to their anti-inflammatory and analgesic activities [11,19,20]. Other studies showed also antioxidant [10,21,22], antimicrobial [23] and anti-leishmanial [1] activities in the *T. catharinensis* extracts. To the best of our knowledge, no other author has reported the antidiabetic nature of different extracts of *T. catharinensis* leaves.

Thus, this study aimed to evaluate the influence of factors such as solvent type, time and temperature on the recovery of individual phenolic compounds from *Tabernaemontana catharinensis* leaves and to evaluate the antioxidant effect and lipid peroxidation in the liver and kidneys of streptozotocin-induced diabetic rats.

## 2. Results and Discussion

The variation of the phenolic compound content is directly related to the independent variables such as the solvent type, time and temperature of the extraction. In fact, each phenolic source demands an individual approach for extraction and optimization [6]. Effect of different solvents, time and temperature on the extraction of individual phenolic compounds from *Tabernaemontana catharinensis* leaves can be seen in Table 1. The chromatographic profile of *Tabernaemontana catharinensis* leaves extract according to factorial design (A1 to A8) demonstrates the presence of quercetin, rutin, ferulic acid, coumaric acid and pinocembrin. The results showed variations in the levels of phenolic compounds and according to Brum et al. [11] the differences in the structure of these compounds also determine their solubility in solvents of different polarity as used in this study.

The highest content of quercetin and rutin (0.26 and 0.51 mg/g, respectively), ferulic acid and coumaric acid (0.09 and 0.054 mg/g respectively) were found in assay A6 and A8, both obtained with ethanol (Table 1). Thus, ethanol can be considered the most suitable solvent for quercetin, rutin, ferulic acid and coumaric acid extraction from *Tabernaemontana catharinensis* leaves when compared to ethyl acetate. The higher solubility of compounds in polar solvent ethanol is mainly due to a sugar portion in the rutin structure and a high degree of OH substituents in quercetin and phenolic acid molecules [24,25].

In addition, the temperature variable and the interaction between temperature and solvent influenced in the ferulic acid content and showed an increase in the A8 assay (Ethanol, 60 min, 65 °C) with significant differences from the others (Table 1). While, the coumaric acid content also increased significantly in the A8 assay (Ethanol, 60 min, 65 °C) when compared to the A2 (Ethanol, 30 min, 35 °C) and A4 (Ethanol, 60 min, 35 °C) assays.

However, when the ethyl acetate solvent has been used, only pinocembrin can be quantified (Table 1). The levels of pinocembrin obtained of extraction using ethyl acetate show significant differences when compared to extracts obtained with ethanol (Table 1), since the ethyl acetate solvent, which has medium polarity, is better for recovering the free aglycones such as pinocembrin (flavonone), flavonols and flavones.

When the extraction temperature increased from 35 to 65 °C, the recovery yields of phenolic compounds significantly increased (Table 1). According to Vedana et al. [26], the use of higher temperatures during the extraction promotes solubility of the compounds and makes the cell walls more permeable. The increased solubility and diffusion of the compounds facilitate the extraction since the solubility of the phenolic compounds is driven by the polarity of the solvent [16]. The solvent type chosen in this study has a polarity similar to the phenolic compounds present in the *T. catharinensis* leaves extract and the extraction results are in agreement with several authors who use polar solvent [10,27,28].

The one-way analysis of variance (ANOVA) showed adequate adjustment of the experimental model data used in this work, which were submitted to the thermal treatment by FD and RSM (Table 2 and Figure 1, Figure 2 and Figure 3). In these experiments, F_value_ for all response variables (phenolic compounds) was always greater than F_statistic table_, and in some cases, this amount was above 185 times greater than F_statistic table_. The lower ratio found in F_value_ by F_statistic table_ in the analysis of variance was 48.10 (Table 2). These results showed that the empirical data were adequately adjusted to the proposed models. The six models generated had correlation coefficients (R^2^) with a variation of 0.94 to 0.97 in which the variability of data is justified by the generated models. ANOVA revealed that the solvent variable had a significant effect (*p* < 0.001) on all responses (Table 2).

The effects of factors (or independent variables), type of solvent, extraction time and temperature were also evaluated by ANOVA (Table 2). The type of solvent was highly significant (*p* < 0.001) over all analyzed responses. This denotes that certain antioxidant compounds are better extracted with a certain type of solvent than with another.

However, the time extraction alone had little effect on the quercetin (*p* < 0.05) and rutin (*p* < 0.01) extraction content yield (Figure 1a–d). The effect of different parameters on rutin yields can be seen in Figure 1c,d. Thus, temperature of 65 °C with ethanol as solvent extractor during 60 min were the best condition to extract the rutin (R^2^ = 0.94) (Table 1, Table 2 and Figure 1c,d). However, these results are not in agreement with those earlier reports that found 5.01 mg/mL of quercetin and 1.10 mg/mL of rutin in the ethyl acetate fraction from *T. catharinensis* leaves [11].

The temperature independent variable not had a significant effect on quercetin, rutin and pinocembrin extraction yields (*p* > 0.05). While, the time extraction not had a significant effect on ferulic acid and pinocembrin extraction yields from *T. catharinensis* leaves (Table 2, Figure 2a,b). Figure 2a,b represent the effect of temperature and solvent on the extraction yield of ferulic acid and pinocembrin, respectively. In this study, a good fit with high correlation (R^2^) was achieved with the regression model for the extraction yields of ferulic acid and pinocembrin, which were 0.95 and 0.97 respectively (Table 2). 

The lowest extraction content for ferulic acid was observed when ethyl acetate was used as solvent, however, this solvent exhibited the highest values of pinocembrin (Table 1, Figure 2a,b). In fact, according to Bitencourt et al. [29] the ferulic acid has greater solubility in ethanol pure and high temperature and is in accordance with our results that have similar extraction characteristics. Furthermore, all the independent variables (solvent, time and temperature) had a significant effect (*p* < 0.001) on coumaric acid extraction yield (Table 2, Figure 3a–c). The best extraction condition of coumaric acid from *T. catharinensis* leaves occurs at a temperature of 65 °C and a time of 60 min using ethanol as solvent (Figure 3a–c).

The global response of the multivariate regression model showed R² = 0.96, implying that 96% of the variation was explained by the model (Table 2). However, some models presented a lack of fit, and the chi-square test for pinocembrin, quercetin, ferulic acid and coumaric acid in the overall response (R^2^ > 0.94, Table 2) was determined to overcome this inadequacy. The analysis indicates that there is no significant difference between the observed and predicted values for these responses and only for the rutin, there was significant difference between the values observed and predicted for the model (Table 2).

In the current study, principal component analysis (PCA) was used to provide more relevant information about the extraction of phenolic compounds from *T. catharinensis* leaves. Briefly, PCA appeared to be a useful tool to report the efficiency of polyphenol extraction and capable of retaining relevant information for to represent each variable individually [30]. The principal component analysis (PCA) was performed on the data set of phenolic compounds profile. Two main components were identified, explaining 98.18% of total variability: PC1 for 87.87% variation and PC2 for 10.31% (Figure 4a,b).

The group formed by coumaric acid and ferulic acid (second quadrant Figure 4a) was strongly influenced by the solvent ethanol, at 65 °C with extraction time of 60 min (A8). The group represented by quercetin and rutin (third quadrant Figure 4a) was shown to be influenced by the solvent ethanol, with heating at 35 and 65 °C and a heating incubation time of 30 and 60 min (A2, A4 and A6). The group represented by pinocembrin (first quadrant Figure 4a) was strongly influenced by the ethyl acetate solvent, heating at 35 and 65 °C and for a time of 30 and 60 min (A1, A3, A5 and A7).

### 2.1. Total Phenolic Compounds and Antioxidant Activity in Vitro

Those results showed that the best extraction condition for phenolic compounds profile was achieved when the ethanol at 65 °C for 60 min was used. In this way, the extract obtained in the A8 run was used to analyze the total phenolic compounds and antioxidant activity for four in vitro methods (2,2-Diphenyl-1-picryl-hydrazyl (DPPH), ferric reducing antioxidant power (FRAP), 2,2-azino-bis-(3-ethylbenzothiazoline-6-sulphonic acid (ABTS) and β-Carotene) (Table 3) and enzymatic activities (catalase (CAT), superoxide dismutase (SOD), protein thiols (PSH)) and lipid peroxidation (LP) of streptozotocin-induced diabetes in male Wistar rats (Figure 5).

*T. catharinensis* extract A8 presented total phenolic compounds of 23.34 mg of EAG/g (Table 3) which is lower than found by Boligon et al. [10] with values ranging from 135.57 to 562.780 mg EAG/g of *T. catharinensis* collected in Bossoroca, Rio Grande do Sul State of Brazil. This difference could be attributed to different solvents and time of extraction used in the experiment. In addition, extraction efficiency is affected by the chemical nature of the compounds, the extraction method used, sample particle size as well as the presence of interferents [10,11,24,25].

The A8 extract of *T. catharinensis* presented a higher level of antioxidant activity regarding free radical scavenging evaluated by DPPH and ABTS methods with values of 34.26 μmol of Trolox and 12.57 µmol Trolox, respectively (Table 3). The extract (A8) had a content of 24.13 μmol Fe^2+^/g by FRAP method and 78.85% by oxidation of βeta-carotene and linoleic acid method (Table 3).

The scarce number of studies on the extraction of chemical compounds of *Tabernaemontana catharinensis* leaves make it difficult to compare them with previous studies, since they are of different species of the genus Tabernaemontana and originate from different regions where the soil and the climate interfere in the results. In addition, the extraction methodologies used in the studies are varied, as there are different factors that can alter the amount of phytochemical compounds present in the plant. However, the *T. catharinensis* presents an excellent source of antioxidant compounds that are capable of acting against possible oxidizing agents [16,21,22].

### 2.2. Biochemical Markers

The metabolic disorder of diabetes, including insulin resistance, hyperglycemia and release of excess free fatty acids, along with other metabolic abnormalities affects tissues of animals. In fact, the endothelial dysfunction, free radicals generated by autoxidation reactions of sugars and sugar adducts in protein has been reported as a common event in the pathogenesis of diabetic complications [31,32]. In addition, the autoxidation of unsaturated lipids in plasma and membrane proteins and finally oxidative stress are increased in diabetic rats [33].

Figure 5 illustrates the effects of lyophilized extract of *T. catharinensis* (TC) on the hepatic antioxidant status in the liver and kidneys of nondiabetic and diabetic rats. A significant interaction between diabetic group and oral administration of 300 mg/Kg of TC in a 30-day treatment period was observed in the liver and kidneys to SOD activities. Liver SOD activity in the untreated diabetic rats was significantly less than the controls (control saline and control TC) and diabetic+TC groups (*p* < 0.05) as shown in Figure 5A,B. The oral administration of 300 mg/Kg of TC extract in a 30-day treatment period restored the normal activity of SOD (*p* < 0.05) to liver and kidney tissues.

A significant interaction between TC extract group and diabetic group was observed for CAT activity in kidney tissue (*p* < 0.05) as shown in Figure 5B. There was a significant decrease (*p* < 0.05) in kidney CAT activity in diabetic rats group compared to the control saline, control TC and diabetic+TC groups (Figure 5B).

The thiobarbituric acid reactive substances (TBARS) levels in kidney tissue (*p* < 0.05) were altered as shown in Figure 5B. In fact, the TBARS levels decreased in kidney tissue in streptozotocin-induced diabetic rats treated with *Tabernaemontana catharinensis* plant when compared to control TC (*p* < 0.05) (Figure 5B). However, CAT activity and TBARS level not were significantly altered (*p* < 0.05) in liver tissue (Figure 5A). The lipid peroxidation increased plays an important role in the progression of diabetes, altering the transbilayer fluidity gradient, which could hamper the activities of the antioxidant enzymatic system [31]. In addition, the study indicates a role of the *Tabernaemontana catharinensis* extract on free radical inactivation and on the antioxidant defense system in diabetic rats, as observed using other natural extract such as *Syzygium cumini* pulp [32], *Spondias tuberosa* inner bark extract [33] and aqueous extract from *Brachylaena elliptica* [34].

A significant decrease in thiol level was observed in the liver and kidneys of diabetic rat (Figure 5A-B). The protein thiols in liver tissue was significantly higher in the control group (saline) than in the diabetic rats as shown in Figure 5A. However, in this tissue, the PSH levels not were significantly altered (*p* < 0.05) in the control saline group when compared to the control TC and diabetic+TC groups. The PSH in the kidneys was significantly decreased in diabetic group when compared with control saline and TC groups (*p* < 0.05) (Figure 5B). Previous reports indicate an increase of intracellular oxidative stress in the development of diabetic complications [35,36,37]. According to Sharifzadeh et al. [38], the significant decrease in thiol content may be due to increased utilization of glutathione (GSH) for scavenging effect in diabetes. Tissue GSH plays a central role in antioxidant defense by detoxifying reactive oxygen species directly or in a glutathione S-transferase (GST) catalyzed mechanism. A decrease in the enzymatic and nonenzymatic antioxidant systems was observed in this present study, in agreement with previous reports in induced diabetic rats [39].

According to Hassan et al. [40], there is evidence that overproduction of free radicals result in oxidative stress. However, many authors have shown that diabetic complications can be reversed when the animals are treated with natural antioxidants [32,33,34,37,41]. Based in our findings, the *T. catharinensis* extract may be able to fight free radicals and modulate SOD and CAT and it increased the scavenging capacity of free radicals in the liver and kidneys of diabetic rats. In addition, the reactivation of SOD activity promoted by *T. catharinensis* may accelerate the dismutation of O_2_^•−^ to H_2_O_2_, which is quickly removed by CAT, protecting the liver and kidneys against free radical damage.

Although, the *T. catharinensis* extract may have other compounds that could not be identified at that time, we can suggest that the rutin, quercetin, ferulic acid, coumaric acid and pinocembrin, probably due to synergism action, they can have contributed for antioxidant activity and beneficial action on biochemical markers in diabetic rats. In fact, these molecules have therapeutic action and were studied previously by several authors. The rutin [42,43], *p*-coumaric acid [44] and ferulic acid [45] have affinity to scavenge free radicals and reactive species, therefore improving the activities of physiological protectors as enzymatic and nonenzymatic antioxidants in diabetic rats. While the quercetin was studied by Roslan et al. [46], who found that this molecule possess beneficial effects in ameliorating diabetic complications as well as to restore cell viability in kidneys of streptozotocin-induced diabetic rats. Pinocembrin has been widely used in the therapy of diseases and due to its antioxidant activities is able to inhibit oxidative stress in cells exposed to advanced glycation end products (AGEs) association with degenerative diseases such as diabetes [47]. Additionally, as suggested by Man et al. [48], seeds of *Litchi chinensis* extract, rich in pinocembrin, improved the quality of life of diabetic rats and protected pancreas, liver and kidney tissues from damage.

Thus, this study is in agreement with others that use natural therapeutic antioxidant compounds that can afford protection in a variety of in vitro and in vivo models of human pathologies, including diabetes models [31,32,33,34]. In fact, the phenolic compounds found in *T. catharinensis* can act directly by entering the redox reactions. Additionally, the results of the present study are consistent with the protective effect of *T. catharinensis* leaves extract reported for the first time, which showed that *T. catharinensis* treatment meliorated antioxidant enzymatic activities and lipid peroxidation in streptozotocin-induced diabetic rats.

## 3. Materials and Methods

### 3.1. Chemicals

Quercetin, rutin, ferulic acid, coumaric acid, pinocembrin, DPPH (2,2-diphenyl-1-picrylhydrazyl), ABTS (2,2′-azino-bis (3-ethylbenzothiazoline-6-sulfphonic acid), TPTZ (2,4,6-tris(2-pyridyl)-s-triazine), Folin–Ciocalteu phenol reagents, Trolox, β-carotene/linoleic acid, 1,2-Bis(dimethylamino)ethane (TEMED), ethylenediamine tetraacetic acid (EDTA) and streptozotocin (STZ) were obtained from Sigma-Aldrich (St. Louis, MO, USA).

### 3.2. Samples

The *Tabernaemontana catharinensis* leaves (TC) were collected in Tenente Portela, Rio Grande do Sul State—Brazil (latitude 27°22’16” S, longitude 53°45’30” W). A dried voucher specimen is preserved in the herbarium of the Department of Biology at Midwestern Parana State University (UNICENTRO, Guarapuava, Brazil) by register number ARAUCA 1084. The plant material was dried in an oven with forced air circulation at 35 °C for 48 h, ground in an analytical mill (Wiley 4, Ramsey, MN, USA) and stored at −12 °C.

### 3.3. Preparation of Extract and Experimental Design

The factorial design (FD) 2³ was used to evaluate the effects of solvent, time and temperature over phenolic compounds extraction. The design was carried out with eight trials and the independent variables were solvent type (X1, ethanol or ethyl acetate), time (X2, 30 min or 60 min) and temperature (X3, 35 °C or 65 °C), on phenolic compounds profile determined by HPLC. Each treatment was performed in three replicates. In this study, the factors choice were based by preliminary experiments and literature [24,25,49,50].

Samples containing 3 g of crushed leaves were extracted separately with 30 mL of each solvent in a shaker (Solab SL222, Piracicaba, Brazil) according to the experimental design shown in Table 1. The extracts were centrifuged at 447× *g* for 15 min (Hermle Z 200 A, Wehingen, Germany) and the supernatants were stored in a freezer at −12 °C until analysis.

### 3.4. HPLC-DAD Profile of Phenolic Compounds

Phenolic acids and flavonoids were performed by using a high performance liquid chromatography (HPLC) system (Varian, 920-LC, Walnut Creek, CA, USA) including an online degasser, an auto-sampler, a column temperature controller, and a diode array detector (DAD). System control and data analysis were processed with Varian HPLC Galaxie Software. Separation was conducted using a C18 RP (5 μm, 250 × 4.6 mm) column, and the temperature was set to 30 °C. The injection volume was 10 μL and the flow rate was 1.0 mL/ min. The mobile phase consisted of a linear gradient of 1% phosphoric acid in water (A) and methanol (B). The gradient started with 30% of B up to 95% of B in 30 min and returned to the initial condition. The total run time was 42 min and the spectral data were collected from 240 to 400 nm. The identification was performed by comparison of retention times and absorption in ultraviolet. The quantification of the phenolic compounds was carried out by external standardization, using a calibration curve in a concentration range varying from 0.5 mg/mL to 60 mg/mL of caffeic acid, ferulic acid, *p*-coumaric acid, rutin, quercetin, pinocembrin, chrysin, kaempferol, mangiferin and galangin. The quantification limit (LQ) and the detection limit (LD) of the overall procedure were 0.35 mg/mL and 0.12 mg/mL and were calculated based on (3 × SD)/s and (10 × SD)/s, respectively, where *s* is the slope of the calibration curves prepared in solvent and SD is the standard deviation of the intercept in the standard curve [49]. The content of phenolic compounds was expressed for each compound in mg/g of sample. The determination of the phenolic compounds by HPLC was done in triplicate.

### 3.5. Total Phenolic Compounds (TPC) and Antioxidant Activities in Vitro

#### 3.5.1. Total Phenolic Compounds (TPC)

The TPC was determined using the Folin-Ciocalteu method described by Singleton et al. [51], and quantified by comparison with a gallic acid standard. 500 µL of the extract at 0.1 g/mL was mixed with 2.5 mL of Folin-Ciocalteu and 2 mL of sodium carbonate 40 g/L. After two hours in darkness at room temperature, the absorbance of the extract was measured at 764 nm in spectrophotometer (UV-VIS Bel Photonics 2000, Piracicaba, Brazil). The results were expressed as mg GAE/g of sample (GAE: gallic acid equivalent, y = 0.0211x − 0.0168, R^2^ = 0.998). All analyses were carried out in triplicate.

#### 3.5.2. DPPH (2,2-Diphenyl-1-picryl-hydrazyl) Radical Scavenging Assay

DPPH free radical scavenging activity was measured as described by Brand-Williams et al. [27]. The reaction medium consisted of 0.5 mL of the extracts, 3.0 mL of ethanol and 0.3 mL of 0.5 mM DPPH solution in ethanol. The mixture was incubated at room temperature in the darkness for 45 min and the absorbance was read using a spectrophotometer (Bel Photonics 2000, Piracicaba, Brazil) at 517 nm. The results were expressed as µmol of Trolox/g of sample and the analyses were carried out in triplicate.

#### 3.5.3. Ferric Reducing Antioxidant Power (FRAP) Assay

The FRAP reagent was prepared with 25 mL of 300 mM acetate buffer (pH 3.6), 2.5 mL of 10 mM TPTZ (2,4,6-tris(2-pyridyl)-s-triazine), in 40 mM HCl, and 2.5 mL of 20 mM FeCl_3_ in aqueous solution [28]. An aliquot of 100 µL of the extract (0.1 g/mL) was added to 3 mL of freshly prepared FRAP reagent. The absorbance was measured in spectrophotometer (Bel Photonics 2000, Piracicaba, Brazil) at 593 nm. Aqueous solutions of ferrous sulphate (3.6 ×10^−3^ to 3.6 × 10^−2^ μmol of ferrous sulfate) were used for calibration (y = 32.345x − 0.0824, R^2^ = 0.998), and the results were expressed as µmol of Fe^+2^/g. All tests were carried out in triplicate.

#### 3.5.4. ABTS (2,2-azino-bis-(3-ethylbenzothiazoline-6-sulphonic acid) Assay

The antioxidant activity by the ABTS method was performed according to Re et al. [52]. The stock solution was produced by reacting 7.4 mM ABTS^°+^ and 2.6 mM potassium persulfate and kept in the dark at room temperature for 16 h before use. The solution was prepared by mixing 1 mL ABTS^°+^ solution with 60 mL ethanol to obtain an absorbance of 0.70 at 734 nm using the spectrophotometer (UV-VIS Bel Photonics 2000, Piracicaba, Brazil). The results were expressed in μmol/g of TEAC (Trolox-equivalent antioxidant capacity, y = 10.666x − 0.5757, R^2^ = 0.999). The assays were carried out in triplicate.

#### 3.5.5. Coupled Oxidation of β-carotene and Linoleic Acid Assay

The measure of antioxidant activity was determined by the coupled oxidation of β-carotene/linoleic acid [53]. Emulsion oxidation was spectrometrically monitored (Bel Photonics 2000, Piracicaba, Brazil) and the absorbance was read at 470 nm, at time zero (*t* = 0) and subsequently after every 20 min, until the characteristic color of β-carotene disappeared in the control reaction (*t* = 100 min). Test samples were evaluated at the final concentration of 0.1 mg/mL. The antioxidant activity was determined as percent inhibition relative to control sample.

### 3.6. Animals

Male Wistar rats (70–90 days; 250–280 g) from the Regional Community University of Chapecó—UNOCHAPECÓ (Chapecó, SC, Brazil) were used in this study. The animals were maintained under controlled standard conditions (12 h light/dark cycle under constant temperature of 23 ± 1 °C) in plastic cages. The animals were fed with commercial feed and water ad libitum for 30 days. All animal procedures were approved by Animal Ethics Committee from Faculty of Pato Branco—FADEP by code number 001/17.

### 3.7. Preparation of Extracts for Animals Diet

The best condition of phenolic compounds extraction was chosen to be used in the diet of streptozotocin-induced diabetes in male Wistar rats. Thus, 30 g of the crushed leaves of *T. catharinensis* were extracted with 300 mL of ethanol in a shaker at 65 °C during 60 min. The solvent was filtered with filter paper, the supernatant was evaporated using rotary evaporator, and the collected extract was freeze-dried and used in the diet of the male Wistar rats through oral gavage.

### 3.8. Experimental Design Animals and Induction of Diabetes

The animals were randomly divided into four groups (six rats per group): control/saline; diabetic; control/*Tabernaemontana catharinensis* 300 mg/Kg (TC); and diabetic + TC 300 mg/kg. *Tabernaemontana catharinensis* extract was diluted in saline solution and administrated via gavage between 4 and 5 p.m. once a day for 30 days, at a volume not exceeding 1 mL/kg.

Type I diabetes was induced by a single intraperitoneal injection of 55 mg/kg streptozotocin diluted in 0.1 M sodium-citrate buffer (pH 4.5). Blood samples were taken from the tail vein 7 days after streptozotocin or vehicle injection to measure glucose levels. Glucose levels were measured with a portable glucometer (ADVANTAGE Boehringer Mannheim, MO, USA). Only animals with fasting glycemia >250 mg/dL were considered diabetic and used in the study. After the treatment, the animals were anesthetized under isoflurane atmosphere and then drawing blood by cardiac puncture. The liver and kidney tissues were removed for biochemical assays.

### 3.9. Biochemical Analysis

Determination of superoxide dismutase (SOD) activity was assayed based on the principle of inhibition of quercetin auto-oxidation by SOD [54]. The reaction solution containing TEMED (0.8 mM), EDTA (0.8 mM) and quercetin (0.014 mM) in phosphate buffer (pH 10.0). Inhibition of quercetin auto-oxidation by SOD was monitored at 406 nm after addition of tissue homogenate. One unit of SOD activity is the amount of enzyme required to bring about 50% inhibition of quercetin auto-oxidation and the results were expressed as units/mg protein.

Catalase activity in tissues was assayed spectrophotometrically (Spectrophotometer U-2001 Hitachi, Tokyo, Japan) by the method of [55], which involves monitoring the disappearance of H_2_O_2_ in the presence of cell homogenate at 240 nm for 90 s.

Tissues protein thiols (PSH) were determined in 40 μL of supernatant and the colorimetric assay was carried out in 1 M phosphate buffer, pH 7.4 as previously described by [56].

Thiobarbituric acid reactive substance (TBARS) levels were determined according to [57] by measuring the concentration of malondialdehyde (MDA) as an end product of lipid peroxidation by reaction with thiobarbituric acid (TBA). The levels of TBARS were expressed as nmol MDA/mg of protein.

### 3.10. Statistical Analysis

The TPC data sets derived from the factorial design (FD) were analyzed by response surface methodology (RSM) and principal component analysis (PCA). The data were processed by one-way analysis of variance (ANOVA). The experiments were performed in duplicate and the results were expressed as mean (*n* = 2). The averages were compared by Tukey’s test at a 95% confidence level using the Statistica 8.0 software (Stat Soft Inc., Tulsa, OK, USA). All data were expressed as mean ± standard error of the mean.

## 4. Conclusions

The effects of temperature, time and type of solvent on phenolic compounds profile were determined by FD, RSM and PCA. The global response was very useful for simplifying and improving the extraction of phenolic compounds. Thus, in this study, the ethanol as an extractor, at a temperature of 65 °C for 60 min was the best extract condition for phenolic compounds from *Tabernaemontana catharinensis* leaves. In the best extraction condition, it was possible to extract phenolic compounds such as phenolic acids (ferulic and coumaric acid) and flavonoids (quercetin, rutin, pinocembrin), which present good antioxidant activity. In addition, these compounds can be responsible for reduction of lipid peroxidation in the liver and kidney tissues of animals. In fact, the antioxidant biomarkers shown alteration in the both tissues in the rats treated with *T. catharinensis*. The diabetic rats treated with *T. catharinensis* extract increased the SOD activity in both tissues, increased the CAT activity in the kidneys, and increased the thiol contents in the liver. Thus, *T. catharinensis* leaves would be considered a source of phenolic compounds that can recover the enzymatic changes in the liver and kidneys provoked by chronic hyperglycemia. This change leads to a series of biochemical events resulting in the production of high levels of free radicals and eventual oxidative stress. Given that phenolic compounds have the ability to modulate enzyme activity and affect the biological responses, it is important to continue to investigate these activities to evaluate the efficacy of this plant as an adjunct to the management of diabetes, since these compounds are indicative of possible new areas of therapy.

## Figures and Tables

**Figure 1 molecules-25-02391-f001:**
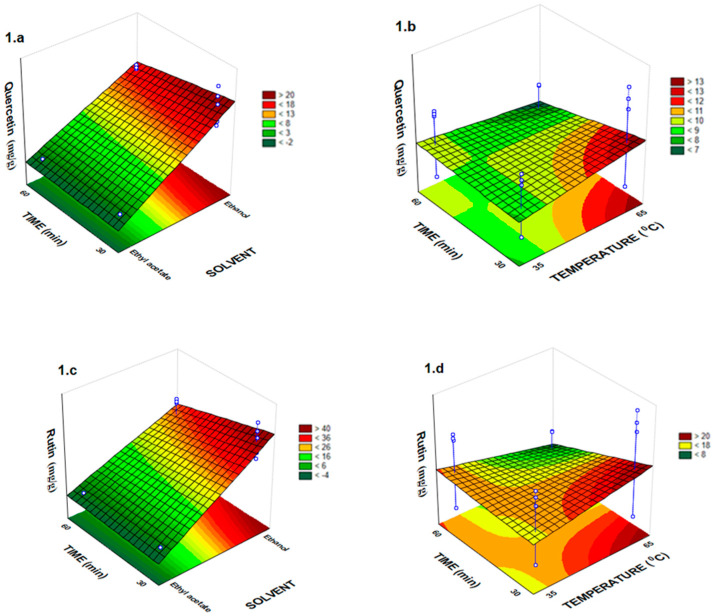
Response surface plots for the effects: time and solvent for the quercetin in (**a**) and rutin in (**c**); time and temperature for the quercetin in (**b**) and rutin in (**d**).

**Figure 2 molecules-25-02391-f002:**
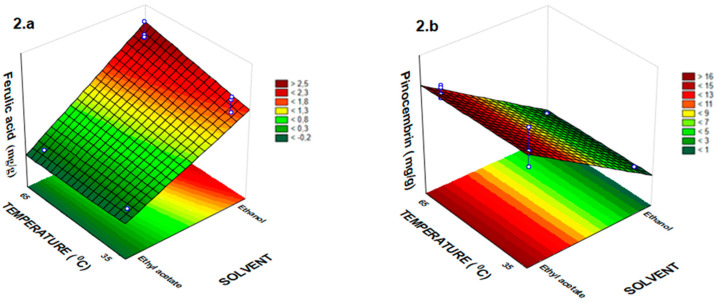
Response surface plots showing the effect temperature and solvent for the ferulic acid in (**a**) and pinocembrin in (**b**).

**Figure 3 molecules-25-02391-f003:**
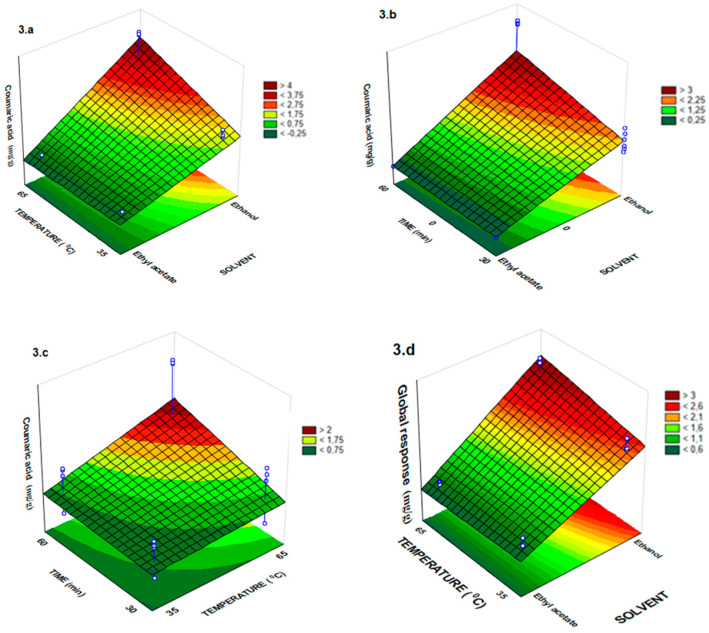
Response surface plots showing the effects of temperature and solvent in (**a**), time and solvent in (**b**), time and temperature in (**c**) for the coumaric acid and global response showing the effect of temperature and solvent in (**d**).

**Figure 4 molecules-25-02391-f004:**
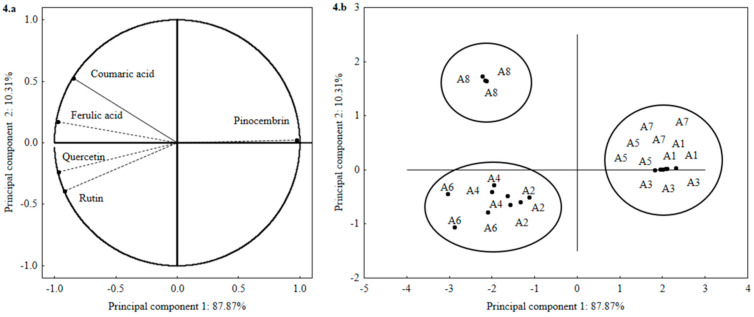
Perceptual map of principal component analysis. Results obtained from distribution of five phenolic compounds in the leaves of *Tabernaemontana catharinensis*. (**a**) Loading plot for the phenolic compounds on principal components 1 and 2; (**b**) scores plot.

**Figure 5 molecules-25-02391-f005:**
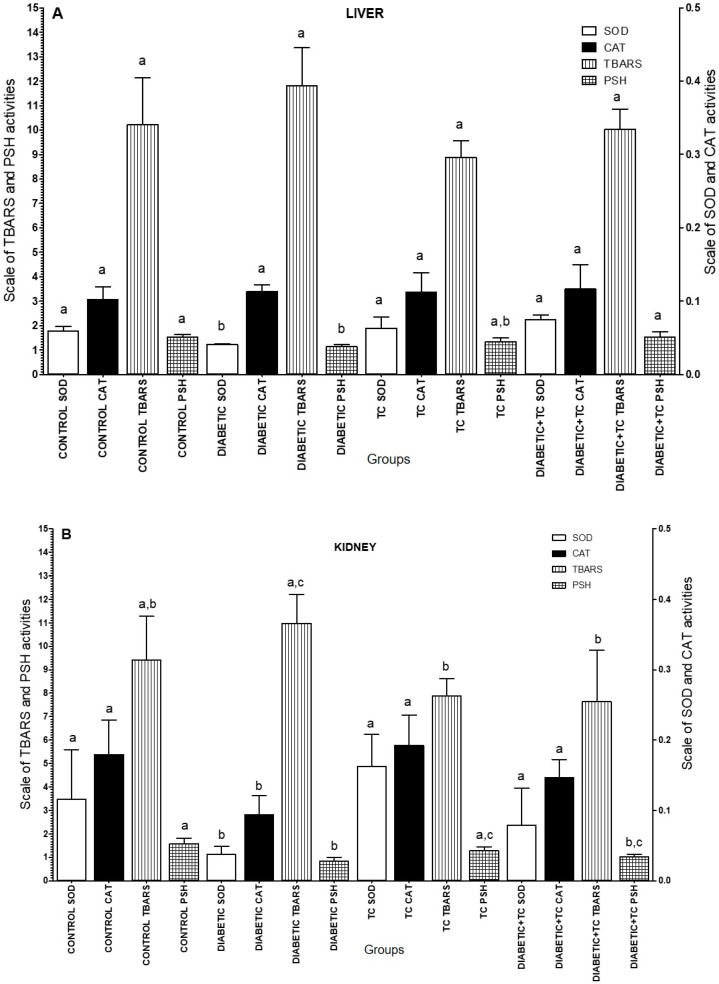
Superoxide dismutase (SOD) and catalase (CAT) activities, thiobarbituric acid reactive substances (TBARS) and protein thiols (PSH) levels in the tissues of liver (**A**) and kidneys (**B**) of streptozotocin-induced diabetic rats treated with *Tabernaemontana catharinensis* (TC) extract. Groups: Control saline (CONTROL), Diabetic (DIABETIC), *Tabernaemontana catharinensis* (TC) 300 mg/Kg, Diabetic+TC (DIABETIC+TC) 300 mg/Kg. Bars represent average standard error for six rats in each group. Different letters (**a**–**c**) at the top of columns indicate significant differences (*p* < 0.05) based on the Tukey’s post-hoc test. The activities were expressed as unit/mg protein for SOD, nmol/mg of protein for CAT, nmol MDA/g tissue for TBARS and as µmol/g tissue for PSH.

**Table 1 molecules-25-02391-t001:** The phenolic compounds content (mg/g) by HPLC of *Tabernaemontana catharinensis* leaves extract.

Assay	Solvent (X1)	Time (min) X2	Temperature (°C) X3	Quercetin	Rutin	Ferulic Acid	Coumaric Acid	Pinocembrin
A1	Ethyl Acetate (−1)	30 (−1)	35 (−1)	<LD	<LD	<LD	<LD	0.15 ^a^ ± 0.01
A2	Ethanol (+1)	30 (−1)	35 (−1)	0.17 ^c^ ± 0.01	0.35 ^c^ ± 0.04	0.02 ^b^ ± 0.00	0.01^c^ ± 0.00	0.02 ^b^ ± 0.00
A3	Ethyl Acetate (−1)	60 (+1)	35 (−1)	<LD	<LD	<LD	<LD	0.15 ^a^ ± 0.01
A4	Ethanol (+1)	60 (+1)	35 (−1)	0.19 ^b^ ± 0.01	0.38 ^b^ ± 0.01	0.02 ^b^ ± 0.00	0.02 ^b,c^ ± 0.00	0.02 ^b^ ± 0.00
A5	Ethyl Acetate (−1)	30 (−1)	65 (+1)	<LD	<LD	<LD	<LD	0.16 ^a^ ± 0.01
A6	Ethanol (+1)	30 (−1)	65 (+1)	0.26 ^a^ ± 0.03	0.51 ^a^ ± 0.04	0.02 ^b^ ± 0.01	0.02 ^b^ ± 0.00	0.02 ^b^ ± 0.00
A7	Ethyl Acetate (−1)	60 (+1)	65 (+1)	<LD	<LD	<LD	<LD	0.14 ^a^ ± 0.01
A8	Ethanol (+1)	60 (+1)	65 (+1)	0.25 ^a^ ± 0.00	0.55 ^a^ ± 0.00	0.09 ^a^± 0.00	0.05 ^a^ ± 0.00	0.04 ^b^ ± 0.00

<LD: Less than limit of detection. Values are presented as mean ± standard deviation (*n* = 3). Different lower-case letter in the same column indicate significant difference (*p* < 0.05) by Tukey’s test.

**Table 2 molecules-25-02391-t002:** Regression coefficients of the models and analysis of variance (ANOVA) for the effect of solvent type (X1), temperature (X2) and time (X3) on phenolic compounds profile.

Source	Quercetin	Rutin	Ferulic Acid	Coumaric Acid	Pinocembrin	Global Response
β_0_	9.38	17.98	1.07	1.37	8.31	1.66
β_1_	9.38	17.98	1.07	1.37	−6.74	0.89
β_2_	0.00	0.00	0.19	0.47	0.00	0.15
β_3_	−1.21	−3.68	0.00	0.39	0.00	0.00
β_12_	0.00	0.00	0.19	0.47	0.00	0.16
β_13_	−1.21	−3.68	0.00	0.39	0.00	0.00
β_23_	−1.70	−4.39	0.00	0.27	0.00	−0.10
R^2^	0.96	0.94	0.95	0.97	0.97	0.96
Main effects						
Solvent (X1)	***	***	***	***	***	***
Temperature (X2)	ns	ns	**	***	ns	**
Time (X3)	*	**	ns	***	ns	ns
F_value_	261.26	342.78	149.12	492.79	757.56	136.63
F*_statistic table_*	2.70	2.77	3.10	2.66	4.30	2.77
F_ratio_	96.76	123.75	48.10	185.26	176.18	49.32
Lack of fit (*p*-value)	0.00	0.00	0.42	0.00	0.64	0.03

*** significant at *p* < 0.001; ** significant at *p* < 0.01; * significant at *p* < 0.05; ns: not significant at *p* > 0.05; F_ratio_: F_value_/F*_statistic table._*

**Table 3 molecules-25-02391-t003:** Total phenolic compounds and antioxidant activity of *Tabernaemontana catharinensis* leaves.

Analysis	Results
TPC DPPH FRAP ABTS β-Carotene/linolenic acid	23.34 mg EAG/g 34.26 µmol Trolox 24.13 µmol Fe^2+^ 12.57 µmol Trolox 78.85 (%)

TPC: Total phenolic compounds, EAG: Equivalent gallic acid.

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
