# Peer review of "Extraction of Phenolic Compounds from Tabernaemontana catharinensis Leaves and Their Effect on Oxidative Stress Markers in Diabetic Rats"

_molecules, 2020, doi:10.3390/molecules25102391_

Round 1

Reviewer 1 Report

Submitted for review article entitled “Optimization of phenolic compounds extraction from Tabernaemontana catharinensis leaves and their effect on oxidative stress markers in diabetic rats” is an original paper. The authors try to  evaluate the most effective extraction condition for recovery of high-value phytochemicals present in the Tabernaemontana catharinensis leaves (TC) and to assess their effect on biochemical parameters in streptozotocin-induced diabetic rats. Currently a lot of research is focused on the natural biological active compounds to minimize toxicological side effect of the chemotherapy in patients. Abstract and Introduction are reasonably clear but all work is a little chaotic. The authors should focus especially on the combination of logical whole identification and then ,,in vivo’’ biological properties. Besides, I think the title is a bit exaggerated. Optimization ?? only on 3 parameters. The methodology is laconic especially for HPLC. Besides, the results are relatively poor with each method. What was the key of action? please explain. The interpretation of the results is correct but I cant find any connections between marked compounds and the discussion. The authors write,, the phenolic compounds found in T. catharinensis can act directly by entering the redox reactions. Additionally, the results of the present study are consistent with the protective effect of T. catharinensis leaves extract reported for the first time, which showed that T. catharinensis treatment meliorated antioxidant enzymatic activities and lipid peroxidation in streptozotocin-induced diabetic rats’’. I suggest to discuss the results more closely and link them to specific compounds- not generally. Besides, it is not known which compounds can be active the more so that the authors wrote earlier about the alkaloids contained in this plant??. Figure 1, 4 and 5 are poor quality- please correct.

Reviewer 2 Report

Dear Authors,

You can see my suggestions and comments in the manuscript.

Best regards

Round 2

Reviewer 1 Report

Tthe authors followed the reviewer's comments.

Reviewer 2 Report

Dear authors,

Review: Extraction of phenolic compounds from Tabernaemontana catharinensis leaves and their effect on oxidative stress markers in diabetic rats.

Some recommendations at the last version of the manuscript, please find you some highlighted text in the PDF file attached:

Line 101: Table 1 Title, please delete the text highlighted

Figures 1, 2, and 3: Please remove the borders in the gray color of each figure. This is not necessary.

Line 195: Please insert a space between letters and symbols highlighted.

Line 208: Please delete the space between symbols highlighted.

Figures 5a and 5b: Please, separate the word: Groups that were added down the axis X of the rest of the names of groups.

I hope that all are safe at home.

Best regards

This manuscript is a resubmission of an earlier submission. The following is a list of the peer review reports and author responses from that submission.

Round 1

Reviewer 1 Report

Tabernaemontana catharinensis is a  medicinal plant, with  documented anti-inflammatory activity, traditionally used to treat skin disorders. The plant contain a diverse assortment of bioactive compounds such as indole alkaloids, flavonoids, phenolic acids, triterpenoids that possess many functional properties e.g. antioxidant, anti-inflammatory, anti-carcinogenic, anti-bacterial. So, optimization of the extraction process of bioactive substances from this plant in the context of the content and antioxidant activity of polyphenols is fully justified and interesting. However, in this study, the scope of research was very limited by the choice of one extraction method and only two solvents (ethanol and ethyl acetate). Meanwhile, it is known that aqueous ethanol solutions are generally more effective eluents for various types of polyphenolic compounds. The typical polyphenolic compounds e.g. acetone, methanol, water should be checked too. Therefore, the results obtained are incomplete and have little cognitive value as well as practical significance and should be supplemented. In my opinion additional experiments are needed.

Author Response

Suggestion accepted.

In fact, aqueous ethanol solutions are generally more effective eluents for various types of phenolic compounds. In addition, our team has been testing different types of solvent extractor, as studied in Casagrande et al. (2018) and De Moura et al. (2018), who used acetone/ ethanol and ethanol/water in the extracts of grape pomace and fruit respectively. Thus, in this study we wanted to use crude extracts to find the most appropriate conditions to extract the phenolic compounds with antioxidant activity and make a phenolic profile scan. We use these two solvents because we wanted to focus on phenolic acids and flavonoids. The ethanol and ethyl acetate were selected as extractor solvents because the extraction efficiency is affected by several factors such as type and concentration of solvent, temperature, time and pH. In addition, the polarity of the solvent and its ability to extract substances by solubilisation.

In this case the presence of various antioxidant compounds with different chemical characteristics and polarities may be soluble in the ethanol and ethyl acetate and they present different polarity index and dielectric constant. Thus, we used ethanol (higher boiling point) and ethyl acetate, which is a little less polar. In addition, both ethyl acetate and ethanol are solvents of low cost and easy access, and can be an economic alternative to the industries for insertion in their productive processes. Ethanol is suitable and safe solvent for plant of compounds is commonly used because is inexpensive, lots of compounds dissolve in it, relatively free of regulation and easily evaporated to be incorporated into food products. Water/ethanol mixtures are possibly the most suitable solvent systems for the extraction of polyphenols due to the different polarities of the bioactive constituents, and the acceptability of this solvent system for human consumption. Furthermore, polyphenols are susceptible to oxidation in high temperature and alkaline environment cause their degradation (Do et al 2014). In this first moment, the objective of the work was the extraction in these conditions. We appreciate the suggestion and may continue this study in the future.

Casagrande, M.; Zanela, J.; Pereira, D.; Lima, V.A.; Oldoni, T.L.C.; Carpes, S.T. Optimization of the extraction of antioxidant phenolic compounds from grape pomace using response surface methodology. J. Food Meas. Charact. 2018, 13, 1120-1129.

De Moura, C.; Reis, A.S.; Silva, L.D.; Lima, V.A.; Oldoni, T.L.C.; Pereira, C.; Carpes, S.T. Optimization of phenolic compounds extraction with antioxidant activity from açaí, blueberry and goji berry using response surface methodology. Emir. J. Food Agric. 2018, 30, 180-189.

Do, Q.D., Angkawijaya, A.E., Tran-Nguyen, P. L., Huynh, L.H., Soetaredjo, F. E., Ismadji, S., Ju, Y-H. (2014). Effect of extraction solvent on total phenol content, total flavonoid content, and antioxidant activity of Limnophila aromatic. Journal of food and drug analysis, 22, 296-302.

Casagrande, M., Zanela, J., Wagner Júnior, A., Busso, C., Wouk, J., Iurckevicz, G., Paula Montanher, P. F., Yamashita, F., Malfatti, C.R.M. (2018). Influence of time, temperature and solvent on the extraction of bioactive compounds of Baccharis dracunculifolia: In vitro antioxidant activity, antimicrobial potential, and phenolic compound quantification. Industrial Crops & Products, 125, 207–219.

Reviewer 2 Report

The authors evaluated the best extraction conditions for phenolic compounds from T. catharinensis by applying a factorial design, multivariate regression model and response surface methodology. In addition, the best extract was analyzed for its antioxidative potential.

The methods are described properly and the results and discussion section is well written and structured. The drawback of the study is its very limited number of parameters tested (2 temperatures, 2 time points and 2 solvents). The study aimed to determine the best extraction conditions, but only few parameters were tested. In addition, the best extraction condition differed depending on the phenolic substance. The alleged best conditions for phenol extraction are not very convincing.

The authors are encouraged to also cite and discuss relevant publications, such as FREE RADICALS AND ANTIOXIDANTS, 2013; 3(2):77-80 and Nat Prod Res. 2018 Aug;32(16):1987-1990.

Line 89: add reference

Line 107/108: indicate how LD and LQ were determined

Line 159: A6 corresponds to 65°C according to table 1.

Unfortunately, there is no line numbering from page 9 onwards.

Conclusion: do you really consider the amounts of bioactives as high, especially as in the discussion many references were cited reporting plant extract with much higher contents of same phenols?

Author Response

We agree with the reviser that the polyphenols have wide spectrum of solubility and others combination of factors, such as concentration of the solvent, temperature, time  and solvent-to-solid ratio may be desirable and effective for extraction. We appreciate the suggestion and may continue this study in the future. In addition, our team has been testing different types of solvent extractor, as studied in Casagrande et al. (2018) and De Moura et al. (2018), who used acetone/ ethanol and ethanol/water in the extracts of grape pomace and fruit respectively. In addition in previous studies with Tabernaemontana, Boligon et al. (2013) concluded that among the samples tested the ethyl acetate fraction showed better activity than others solvents.

Regarding the inclusion of a new reference, we appreciate the opportunity to improve our work by including these important studies. Thanks very much. Please see the new version of the manuscript.

Line 89: add reference

Thanks for the suggestion. Please see the new version of the manuscript and the reference was included in the sentence…..The factors choice were based by preliminary experiments and literature [36-39].

Oldoni, T.L.C.; Oliveira, S.C.; Andolfato, S.; Karling, M.; Calegari, M.A.; Sado, R.Y.; Maia, F.M.; Alencar, S.M.; Lima, V.A. Chemical characterization and optimization of the extraction process of bioactive compounds from propolis produced by selected bees. J. Braz. Chem. Soc. 2015, 26, 2054–2062.

Casagrande, M.; Zanela, J.; Pereira, D.; Lima, V.A.; Oldoni, T.L.C.; Carpes, S.T. Optimization of the extraction of antioxidant phenolic compounds from grape pomace using response surface methodology. J. Food Meas. Charact. 2018, 13, 1120-1129.

De Moura, C.; Reis, A.S.; Silva, L.D.; Lima, V.A.; Oldoni, T.L.C.; Pereira, C.; Carpes, S.T. Optimization of phenolic compounds extraction with antioxidant activity from açaí, blueberry and goji berry using response surface methodology. Emir. J. Food Agric. 2018, 30, 180-189.

Boligon, A.A.; Freitas, R.B.; Brum, T.F.; Piana, M.; Belke, B.V.; Rocha, J.B.T.; Athayde, M.L. Phytochemical constituents and in vitro antioxidant capacity of Tabernaemontana catharinensis A. DC. Free Radic. Antioxidant 2013, 3, 77-80.

Pauleti, N.N.; Mello, J.; Siebert, D.A.; Micke, G.A.; de Albuquerque, C.A.C.; Alberton, M.D.; Barauna, S.C. Characterisation of phenolic compounds of the ethyl acetate fraction from Tabernaemontana catharinensis and its potential antidepressant-like effect. Nat Prod Res. 2018, 32, 1987-1990.

Line 107/108: indicate how LD and LQ were determined…

Thanks for suggestion, and was included other information in the methodology.

The limits of detection and quantification were calculated by the following formula:

Where, LD = Limit of detection, LQ = limit of quantification, SD = standard deviation, b = slope of the straight line

Reviewer 3 Report

This study investigated the optimization of extraction method for extracting antioxidant polyphenols from Tabernaemontana catharinensis leaves extract. Overall, the manuscript is well written, but its novelty is limited due to a lot of similar published papers, just with different plant samples. Below are several specific comments.

Please state the difference of this paper from previous similar optimization papers. The Tabernaemontana catharinensis leaves should be authenticated by experts. Line 111: TFC.....TPC Line 148: Please indicate how many times of experiments were repeated. Figures 1 - 3 are not clear. Table 4, the font is not correct, and the table is not complete. There are still some errors in the manuscript, please carefully check them.

Author Response

Thanks for your suggestion. Please see the new version of manuscript

Line 111: TFC.....TPC

We apologize for our mistake. The new version was rewritten.

Line 148: Please indicate how many times of experiments were repeated.

Thanks for your correction and new text was inserted in the methodology.

….. The experiments were performed in duplicate and the results were expressed as mean (n=2)… Please see the new version of the manuscript.

Figures 1 - 3 are not clear.

I believe that after editing they get bigger and more visible.

Table 4, the font is not correct, and the table is not complete. There are still some errors in the manuscript, please carefully check them.

Thanks for your correction and new table was inserted.

Round 2

Reviewer 2 Report

The authors partially improved the manuscript by considering some minor suggestions of the reviewer. However, the main drawback of the study could not be overcome. Only limited parameters were examined which do not allow to identify the best extraction condition as claimed by the authors.

The authors seem to go along with the reviewer's opinion by suggesting to continue this study in the future.

However, the reviewer does not believe that these limited data are enough for an original contribution.

Reviewer 3 Report

The authors only replied to minor comments, but bypass my critical comments. Therefore, the overall quality of the manuscript has not been evidently improved.